# First-line antiretroviral therapy initiation for newly diagnosed people with HIV in the Netherlands: A retrospective analysis from 2016 to 2020

**Piter Oosterhof**[1,2]*, **Ferdinand W. N. M. Wit**[3,4,5], **Matthijs van Luin**[6], **Marc van der Valk**[3,4,5], **Kees Brinkman**[7], **David M. Burger**[2]

1 Department of Clinical Pharmacy, OLVG Hospital, Amsterdam, the Netherlands, 2 Department of Pharmacy, Radboudumc Research Institute for Medical Innovation (RIMI), Radboud University Medical Center, Nijmegen, The Netherlands, 3 Division of Infectious Diseases, Amsterdam University Medical Centers, Location Academic Medical Center, Department of Internal Medicine, Amsterdam Institute for Infection and Immunity, University of Amsterdam, Amsterdam, The Netherlands, 4 HIV Monitoring Foundation, Amsterdam, Netherlands, 5 Department of Infectious Diseases, Public Health Service of Amsterdam, Amsterdam, Netherlands, 6 Department of Clinical Pharmacy, Meander Medical Center, Amersfoort, the Netherlands, 7 Division of Infectious Diseases, Department of Internal Medicine, OLVG, Amsterdam, The Netherlands

* piter.oosterhof@radboudumc.nl

## Abstract

### Introduction

HIV treating physicians in the Netherlands follow the guidelines of the Department of Health and Human Services (DHHS). Most of these recommended initial regimens are single-tablet regimens (STRs), which incur higher costs. By the end of 2017, generic NRTI backbones had become widely available, offering a potentially cheaper multi-tablet regimen. This study aimed to evaluate guideline compliance in people with HIV who started antiretroviral therapy (ART), the uptake of generic multi-tablet regimens (gMTRs), and associated medication costs.

### Methods

This retrospective cohort study used data from the Dutch HIV Monitoring Foundation to determine the proportion of treatment-naïve people entering care who initiated ART according to the DHHS and type of ART regimens prescribed between January 2016 and December 2020. We analyzed ART prescriptions, both at the national level and per individual HIV treatment centers. We calculated the monthly ART costs based on Dutch medicine prices listed on www.medicijnkosten.nl for each calendar year.

### Results

In 2016, an integrase inhibitor-containing regimen was initiated in 77.3% which increased to 87.8% in 2020. The compliance rate to DHHS-recommended initial regimens ranged from 82.8% in 2016 to 90.9% in 2020. Most patients received single-tablet regimens, 81.3% in

**Data Availability Statement:** The datasets utilized in this study are held by Stichting HIV Monitoring

(SHM) and are available upon request. Researchers wishing to access the SHM datasets must submit a request form, which is detailed on the SHM website (https://www.hiv-monitoring.nl/en). The form should be sent to hiv.monitoring@amsterdamumc.nl, accompanied by any supporting documents in Word or PDF format. Approval of data access requests is contingent upon review and endorsement by SHM's working group and governing board. Following approval, a data usage agreement must be completed and submitted to SHM. The authors did not have any special access privileges to this data that other researchers would not have.

**Funding:** The author(s) received no specific funding for this work.

**Competing interests:** PO received personal consulting fees from ViiV Healthcare, Gilead, Merck Sharp, and Dohme. FW has been served on scientific advisory boards for ViiV Healthcare and Gilead Sciences. MvdV has served on advisory boards for Gilead Sciences (Merck, Sharp, & Dohme) and ViiV Healthcare. MvdV received independent scientific grant support from Gilead Sciences, Merck, Sharp, & Dohme, and ViiV Healthcare, all of which were paid to his institution (Amsterdam University Medical Centers, Amsterdam, Netherlands). KB received personal consultation fees from Merck, Gilead, and ViiV. KB has received a writing honorarium from UpToDate. The remaining authors declare that they have no conflicts of interest." I confirm that this not alter my adherence to all PLOS ONE policies. And thereby: This does not alter our adherence to PLOS ONE policies on sharing data and materials.

**Abbreviations:** 3TC, lamivudine; ABC, abacavir; ABC/3TC, abacavir/lamivudine; ART, antiretroviral therapy; ATV, atazanavir; BIC, bictegravir; bMTRs, branded MTRs; DHHS, Department of Health and Human Services; DOR, doravirine; DRV, darunavir; DRV/c, darunavir/cobicistat; DRV/c/TAF/FTC, darunavir/cobicistat/tenofovir alafenamide fumarate/emtricitabine; DTG, dolutegravir; EFV, efavirenz; EVG/c, elvitegravir/cobicistat; FTC, emtricitabine; gMTRs, generic multi-tablet regimens; HKZ—Dutch, Harmonisatie Kwaliteitsbeoordeling in de Zorgsector; INSTI, integrase inhibitor; MTRs, multi-tablet regimens; NVHB, Dutch Association of HIV Treatment Physicians; RAL, raltegravir; RPV, rilpivirine; RPV/TAF/FTC, rilpivirine/tenofovir alafenamide fumarate/emtricitabine; RTV, ritonavir; SFK, Foundation for Pharmaceutical Statistics; SHM—Dutch, Stichting HIV Monitoring; STRs, single-tablet regimens; TAF, tenofovir alafenamide; TAF/FTC, tenofovir alafenamide fumarate/emtricitabine; TDF, tenofovir

2016 to 60.3% in 2020. After the introduction the gMTRs showed a steady increase from 17.8% in 2018 to 37.8% in 2020. The cost of the first-line regimen per patient decreased by 22.9% in 2020 compared with 2017. The decrease was larger in centers where treatment-naïve individuals with HIV were preferentially initiated on a gMTR.

## Conclusions

There was a high compliance to the "DHHS-recommended initial regimens for most people with HIV" in the Netherlands. Most people who initiated ART received STRs, although the percentage of people who started on STRs gradually decreased over time. The use of gMTRs increased over time and was associated with lower medication costs.

## Introduction

In the Netherlands, the HIV Treatment Guideline from the Dutch Association of HIV Treatment Physicians (NVHB) follow the guidelines of the U.S.A. Department of Health and Human Services (DHHS) for the choice of first-line antiretroviral therapy (ART) [1]. Recent years have shown a shift in these guidelines towards recommending starting with integrase inhibitor (INSTI) containing regimens, based on their high efficacy and good tolerability [2]. Most of these regimens come as a single-tablet regimens (STRs) [3]. The existing literature suggests that DHHS-recommended STRs are associated with better adherence and improved clinical outcomes but they are also associated with higher medication costs [4–6].

The cost of ART for the treatment for people with HIV is a significant burden for healthcare systems worldwide [7, 8]. In recent years, the increased use of certain new combination therapies in a STR has substantially increased the cost of HIV treatment [9, 10]. Generic NRTI backbones (tenofovir disoproxil fumarate/emtricitabine (TDF/FTC) and abacavir/lamivudine (ABC/3TC)) were introduced in the Netherlands by the end of 2017. Generic multi-tablet regimens (gMTRs) are clearly less expensive than STRs, with an average cost reduction of 25% [11–13].

The DHHS guideline committee suggests that prescribers select a recommended regimen for a particular individual on "virologic efficacy, toxicity, pill burden, dosing frequency, drug—drug interaction potential, resistance test results, comorbid conditions, access, and cost" [3]. It can be stated that the recommended starting regimens do not differ greatly in all of these aspects, with the exception of pill burden and costs. In 2018, the DHHS guidelines recommended multi-tablet regimens including a generic NRTI backbone (gMTR) in addition to branded STRs [3]. The selection of gMTR as a first-line ART regimen could significantly lower medication costs while preserving high virological efficacy and good tolerability [14].

We conducted a retrospective cohort study of first-line regimens prescribed in people with HIV in the Netherlands between 2016 and 2020. Our aim was to evaluate guideline compliance, gMTR uptake, and associated costs for people starting ART. In addition, the variability between treatment centers was investigated.

## Methods

### Study population

We conducted a retrospective cohort study using pseudonomized data from the Dutch HIV Monitoring Foundation (in Dutch, Stichting HIV Monitoring [SHM]). This is a nationwide

disoproxil fumarate; TDF/FTC, tenofovir disoproxil fumarate/emtricitabine.

registry of all HIV treatment centers in which all people with HIV who do not opt out are registered [15]. The study period was from January 2016 to December 2020. We included all treatment-naïve people with HIV who started first-line ART during this period.

## DHHS guideline

We reviewed "the recommended initial regimens for most people with HIV" by the DHHS guideline committee for each calendar year of the study period, as listed in Table 1 [3]. To clarify, we did not conduct analyses for the "recommended initial regimens in certain clinical situations".

For each calendar year, we defined a regimen as "DHHS recommended" based on the version of the guidelines in effect on January 1st. In other words, if a regimen was removed from the DHHS list in an update of the DHHS guidelines released later in the same calendar year, we considered the regimen as "DHHS recommended" for the rest of the calendar year.

## Data analysis and statistics

We analyzed the prescribed first-line regimens of people with HIV aged 18 years and older and calculated the percentage of patients that started a DHHS-recommended first-line regimen, only for the first monthly prescription. Subsequent analysis of the prescribed DHHS-recommended regimens included the proportion of STRs versus multi-tablet regimens (MTRs). The MTRs were further classified into branded MTRs (bMTRs) and gMTRs. bMTRs were branded dolutegravir or raltegravir, both in combination with a branded backbone of tenofovir alafenamide fumarate/emtricitabine (TAF/FTC) or tenofovir disoproxil fumarate/

**Table 1. DHHS-recommended initial regimens for most people with HIV per year with monthly prices.**

| Regimen | Year 2016[a] cost (€) | 2017[b] cost (€) | 2018[c] cost (€) | 2019[d] cost (€) | 2020[e] cost (€) |
|---|---|---|---|---|---|
| **Single-tablet regimens** | | | | | |
| BIC/TAF/FTC | | | 899.00 | 899.00 | 872.95 |
| DTG/ABC/3TC | 899.00 | 899.00 | 899.00 | 899.00 | 886.20 |
| EVG/c/TAF/FTC | 921.00 | 921.00 | 921.00 | | |
| EVG/c/TDF/FTC | 969.50 | 969.50 | 969.50 | | |
| **Multi-tablet regimens** | | | | | |
| DTG, TDF/FTC | 1,118.80 | 1,087.16 | 610.59† | 610.59† | 609.90† |
| RAL, TDF/FTC | 1,172.77 | 1,103.53 | 621.15† | 619.42† | 624.25† |
| DRV/RTV, TDF/FTC | 1,109.10 | | | | |
| DTG, TAF/FTC | | 1,085.58 | 1,085.58 | 1,081.50 | 1,068.74 |
| RAL, TAF/FTC | | 1,101.95 | 1,096.14 | 1,090.33 | 1,083.09 |

[a] DHHS updated on July 14, 2016; medicine prices are from April 2016

[b] DHHS updated on October 17, 2017; medicine prices are from April 2017

[c] DHHS updated on October 25, 2018; medicine prices are from April 2018

[d] DHHS updated on December 18, 2019; medicine prices are from April 2019

[e] DHHS updated on June 3, 2020; medicine prices are from April 2020

† calculated as a combination antiretroviral therapy containing a generic backbone (TDF/FTC), since introduction in 2017

3TC: lamivudine, ABC: abacavir, BIC: bictegravir, DRV: darunavir, DTG: dolutegravir, EVG/c: elvitegravir/cobicistat; FTC: emtricitabine; RAL: raltegravir; RTV: ritonavir; TAF: tenofovir alafenamide; TDF: tenofovir disoproxil fumarate

emtricitabine (TDF/FTC). gMTRs were branded dolutegravir or raltegravir, both in combination with a generic backbone of TDF/FTC available since 2018.

The analysis of costs associated with gMTR versus STR/bMTR was restricted to 2017–2020 because gMTR was not available before this time period. Although the SHM does not record whether a formulation is branded or generic, we assumed that pharmacies dispensed generic formulations because in the Netherlands, once a generic formulation is available, the branded formulation is no longer reimbursed, with only a few exceptions for individual patients, which we disregarded.

We calculated the monthly ART costs based on Dutch medicine prices listed on www. medicijnkosten.nl, based on the April price for each calendar year [16]. This choice was made because medicine prices in the Netherlands are subject to change twice a year: in April and October. By using the April price, we ensured that our calculations incorporated the most up-to-date and current prices throughout the calendar year. The prices of the drugs are listed in Table 1. We performed the analysis at both at the national level and at the level of individual treatment centers.

Descriptive statistics, including mean, standard deviation, median, and interquartile range, were used to summarize the data. We did not perform any statistical tests or models for hypothesis testing or association analyses, as we considered this exploratory analysis.

### Ethics

We maintained the SHM database according to the Dutch Data Protection Act and Medical Treatment Contracts Act. Given that data collection is an inherent component of the SHM, individuals under HIV care are provided with written materials and informed verbally by their attending physician about the purpose of data collection, with the option to provide verbal consent or opt out. Since our study involved analysising pre-existing data and the data collection process aligns with the routine operations of the SHM, we deemed it exempt from additional ethical approval requirements. We coded the treatment centers to ensure investigators could not identify them. Additionally, we pseudonymized data from each treatment center to protect privacy and confidentiality while maintaining traceability for analysis per calendar year.

## Results

### Initial regimens

We included 3,445 treatment-naïve people with HIV who initiated first-line ART in the Netherlands between January 2016 and December 2020. Table 2 presents the detailed information on the initial regimens prescribed during this period. In 2016, 77.3% started an INSTI-containing regimen, which increased to 87.8% by 2020. In 2018, there was a temporary decrease in the number of INSTI-based regimens prescribed, coinciding with an increase in prescriptions for the newly introduced single-tablet regimens rilpivirine/tenofovir alafenamide fumarate/emtricitabine (RPV/TAF/FTC) and darunavir/cobicistat/tenofovir alafenamide fumarate/emtricitabine (DRV/c/TAF/FTC).

The prescriptions for specific regimens showed varying trends over time. There was a decline in prescriptions for the STRs dolutegravir/abacavir/lamivudine, elvitegravir/cobicistat/tenofovir alafenamide fumarate/emtricitabine, and elvitegravir/cobicistat/tenofovir disoproxil fumarate/emtricitabine. Conversely, there was an increase in prescriptions for the STR bictegravir/tenofovir alafenamide fumarate/emtricitabine (BIC/TAF/FTC), and the multi-tablet regimen dolutegravir plus TDF/FTC (DTG, TDF/FTC).

**Table 2. First-line regimens for people with HIV in the Netherlands from 2016–2020.**

| Regimen† | Year<br>2016<br>n (%) | 2017<br>n (%) | 2018<br>n (%) | 2019<br>n (%) | 2020<br>n (%) |
|---|---|---|---|---|---|
| **Integrase-inhibitor (n, %)** | 724 (77.3) | 697 (83.7) | 554 (77.9) | 507 (85.4) | 325 (87.8) |
| BIC/TAF/FTC | 0 (0.0) | 0 (0.0) | 54 (7.6) | 308 (51.9) | 177 (47.8) |
| DTG/ABC/3TC | 353 (37.7) | 279 (33.5) | 175 (24.6) | 44 (7.4) | 16 (4.3) |
| DTG, TDF/FTC | 96 (10.2) | 72 (8.6) | 78 (11.0) | 115 (19.4) | 119 (32.2) |
| DTG, TAF/FTC | 6 (0.6) | 54 (6.5) | 40 (5.6) | 13 (2.2) | 5 (1.4) |
| EVG/c/TAF/FTC | 187 (20.0) | 237 (28.5) | 173 (24.3) | 15 (2.5) | 5 (1.4) |
| EVG/c/TDF/FTC | 78 (8.3) | 45 (5.4) | 14 (2.0) | 4 (0.7) | 0 (0.0) |
| RAL, TDF/FTC | 3 (0.3) | 4 (0.5) | 12 (1.7) | 7 (1.2) | 2 (0.5) |
| RAL, TAF/FTC | 1 (0.1) | 6 (0.7) | 8 (1.1) | 1 (0.2) | 1 (0.3) |
| **Protease inhibitors (n, %)** | 74 (7.9) | 62 (7.4) | 70 (9.8) | 51 (8.6) | 12 (3.2) |
| DRV/c/TAF/FTC | 0 (0.0) | 3 (0.4) | 55 (7.7) | 36 (6.1) | 6 (1.6) |
| DRV/RTV, TDF/FTC | 43 (4.6) | 26 (3.1) | 8 (1.1) | 10 (1.7) | 5 (1.6) |
| DRV/c, TAF/FTC | 18 (1.9) | 7 (0.8) | 2 (0.3) | 1 (0.2) | 1 (0.3) |
| ATV, RTV, TDF/FTC | 13 (1.4) | 3 (0.4) | 5 (0.7) | 4 (0.7) | 0 (0.0) |
| DRV/c, TAF/FTC | 0 (0.0) | 23 (2.8) | 0 (0.0) | 0 (0.0) | 0 (0.0) |
| NNRTIs‡ (n, %) | 66 (7.0) | 33 (4.0) | 44 (6.2) | 12 (2.0) | 18 (4.9) |
| RPV/TAF/FTC | 4 (0.4) | 17 (2.0) | 35 (4.9) | 4 (0.7) | 1 (0.3) |
| EFV/TDF/FTC | 38 (4.1) | 9 (1.1) | 9 (1.3) | 3 (0.5) | 1 (0.3) |
| RPV/TDF/FTC | 24 (2.6) | 7 (0.8) | 0 (0.0) | 1 (0.2) | 0 (0.0) |
| DOR/TDF/3TC | 0 (0.0) | 0 (0.0) | 0 (0.0) | 4 (0.7) | 16 (4.3) |
| **Miscellaneous (n, %)** | 73 (7.8) | 41 (4.9) | 43 (6.0) | 24 (4.0) | 15 (4.1) |
| DTG, DRV/c, TDF/FTC | 18 (1.9) | 0 (0.0) | 6 (0.8) | 0 (0.0) | 2 (0.5) |
| DTG, DRV/RTV, TDF/FTC | 4 (0.4) | 5 (0.6) | 5 (0.7) | 1 (0.2) | 5 (1.4) |
| Other | 51 (5.4) | 36 (4.3) | 32 (4.5) | 23 (3.9) | 8 (2.2) |
| **Total** | 937 | 833 | 711 | 594 | 370 |

† regimens are listed if they were prescribed in at least 1,0% of the population in any year

‡ non-nucleoside reverse transcriptase inhibitor

3TC: lamivudine, ABC: abacavir, ATV: atazanavir, BIC: bictegravir, DOR: doravirine, DRV: darunavir, DRV/c: darunavir/cobicistat, DTG: dolutegravir, EFV: efavirenz, EVG/c: elvitegravir/cobicistat; FTC: emtricitabine; RAL: raltegravir; RPV: rilpivirine; RTV: ritonavir; TAF: tenofovir alafenamide; TDF: tenofovir disoproxil fumarate

## Guideline compliance

At the national level, the proportion of patients who started a DHHS "recommended initial regimens for most people with HIV" between 2016 and 2020 was between 80–90%, with the exception of 2018, when it was lower (77.9%) (Table 3). The latter can be partly attributed to a temporary increase in prescriptions for regimens that were never included in the DHHS "recommended initial regimens for most people with HIV" (i.e., DRV/c/TAF/FTC and RPV/TAF/FTC).

Table 4 shows the percentage of participants from each center who started the DHHS-recommended initial regimen each year from 2016 to 2020. With a few exceptions, guideline compliance was high (arbitrarily defined as >75%) in all treatment centers during the follow-up period. Three out of the 27 treatment centers had a guideline compliance of <75% in three or more years.

**Table 3.** Patients on DHHS-recommended initial regimens from 2016–2020.

| Regimen | Year 2016 n | 2017 n | 2018 n | 2019 n | 2020 n |
|---|---|---|---|---|---|
| **Single-tablet regimens (n, %)** | 618 (81.3) | 561 (80.5) | 416 (75.1) | 352 (72.1) | 193 (60.3) |
| BIC/TAF/FTC | | | 54 (9.7) | 308 (63.1) | 177 (55.3) |
| DTG/ABC/3TC | 353 (46.4) | 279 (40.0) | 175 (31.6) | 44 (9.0) | 16 (5.0) |
| EVG/c/TAF/FTC | 187 (24.6) | 237 (34.0) | 173 (31.2) | | |
| EVG/c/TDF/FTC | 78 (10.3) | 45 (6.5) | 14 (2.5) | | |
| **Branded multi-tablet regimens (n, %)** | 99 (13.0) | 136 (19.5) | 48 (8.7) | 14 (2.9) | 6 (1.9) |
| DTG, TDF/FTC | 96 (12.6) | 72 (10.3) | | | |
| RAL, TDF/FTC | 3 (0.4) | 4 (0.6) | | | |
| DTG, TAF/FTC | | 54 (7.7) | 40 (7.2) | 13 (2.7) | 5 (1.6) |
| RAL, TAF/FTC | | 6 (0.9) | 8 (1.4) | 1 (0.2) | 1 (0.3) |
| **Generic multi-tablet regimens† (n, %)** | - | - | 90 (16.2) | 122 (25.0) | 121 (37.8) |
| DTG, TDF/FTC | | | 78 (14.1) | 115 (23.6) | 119 (37.2) |
| RAL, TDF/FTC | | | 12 (2.2) | 7 (1.4) | 2 (0.6) |
| **Other regimens (n, %)** | 43 (5.7) | | | | |
| DRV/RTV, TDF/FTC | 43 | | | | |
| **Total** | 760 | 697 | 554 | 488 | 320 |

† Calculated as antiretroviral therapy containing a generic backbone (TDF/FTC), introduction in 2017

3TC: lamivudine, ABC: abacavir, BIC: bictegravir, DRV: darunavir, DTG: dolutegravir, EVG/c: elvitegravir/cobicistat, FTC: emtricitabine, RAL: raltegravir, RTV: ritonavir, TAF: tenofovir alafenamide, TDF: tenofovir disoproxil fumarate

## gMTR uptake

At the national level starting a DHHS-recommended initial regimen, the majority of people with HIV received STRs, with a gradual decrease from 81.3% in 2016 to 60.3% in 2020 (Table 3). Generic MTRs showed a steady increase over the years, reaching 37.8% by 2020 (Fig 1 and Table 3).

Among the DHHS-recommended initial regimens, the use of gMTRs at different treatment centers since 2018 is shown in Fig 2 (S1 Table). A large variability in the uptake of gMTRs as a first-line ART regimen was observed between treatment centers, with 8/25 treatment centers not using gMTRs in 2020 versus 5/25 treatment centers using gMTRs in >75% of their people with HIV initiating ART in the same year.

## Medication costs

The total expenditure on ART regimens decreased from €719,938 in 2016 to €248,944 in 2020. In the last two years, the regimens BIC/TAF/FTC and generic DTG, TDF/FTC had the largest share of the costs (Table 5). The average monthly medication costs per treatment-naïve people with HIV who initiated DHHS-recommended ART were relatively high in 2017, at 950 per month, and decreased to €732 per month by 2020 (Table 6). As this table also shows, there was considerable variation in medication costs between treatment centers, ranging from €600 to >€ 1000 during the five-year study period.

Centers with a higher proportion of people with HIV initiating gMTRs had lower medication costs per person per month for initiating ART (Fig 2). For example, at the extremes, in 2020 the medication costs of the initial treatment regime per person in care in the treatment

**Table 4. Compliance rate on DHHS-recommended initial regimens per treatment center from 2016–2020.**

| Center | Year 2016 n (%) | 2017 n (%) | 2018 n (%) | 2019 n (%) | 2020 n (%) |
|---|---|---|---|---|---|
| 1 | 65 (78.3) | 70 (81.4) | 58 (65.9) | 71 (91.0) | 44 (89.8) |
| 2 | 86 (77.5) | 86 (96.6) | 51 (82.3) | 57 (85.1) | 35 (92.1) |
| 3 | 45 (81.8) | 35 (52.2) | 18 (34.6) | 33 (64.7) | 20 (90.9) |
| 4 | 42 (82.4) | 54 (85.7) | 53 (89.8) | 34 (100.0) | 24 (75.0) |
| 5 | 53 (86.9) | 37 (86.0) | 35 (83.3) | 29 (74.4) | 26 (86.7) |
| 6 | 33 (62.3) | 43 (91.5) | 45 (81.8) | 27 (75.0) | 18 (85.7) |
| 7 | 52 (86.7) | 41 (91.1) | 31 (88.6) | 23 (88.5) | 20 (100.0) |
| 8 | 31 (81.6) | 43 (86.0) | 34 (91.9) | 23 (100.0) | 9 (75.0) |
| 9 | 32 (84.2) | 28 (93.3) | 13 (86.7) | 25 (92.6) | 20 (87.0) |
| 10 | 24 (82.8) | 24 (63.2) | 21 (72.4) | 8 (44.4) | 10 (76.9) |
| 11 | 28 (87.5) | 21 (87.5) | 24 (85.7) | 21 (87.5) | 10 (100.0) |
| 12 | 26 (83.9) | 33 (86.8) | 11 (57.9) | 19 (90.5) | 4 (66.7) |
| 13 | 29 (96.7) | 19 (95.0) | 24 (92.3) | 16 (94.1) | 13 (92.9) |
| 14 | 18 (94.7) | 34 (97.1) | 17 (100.0) | 20 (83.3) | 9 (100.0) |
| 15 | 27 (93.1) | 23 (92.0) | 24 (92.3) | 12 (100) | 8 (100.0) |
| 16 | 21 (80.8) | 16 (69.6) | 18 (78.3) | 12 (92.3) | 7 (100.0) |
| 17 | 20 (87.0) | 13 (81.3) | 6 (60.0) | 8 (33.3) | 5 (62.5) |
| 18 | 14 (93.3) | 11 (84.6) | 17 (89.5) | 8 (80.0) | 10 (90.9) |
| 19 | 17 (54.8) | 6 (54.5) | 9 (90.0) | 4 (80.0) | 3 (100.0) |
| 20 | 15 (87.0) | 10 (83.3) | 3 (30.0) | 8 (100.0) | 9 (100.0) |
| 21 | 10 (83.3) | 6 (85.7) | 13 (86.7) | 11 (91.7) | 5 (55.6) |
| 22 | 10 (100.0) | 11 (100.0) | 8 (80.0) | 6 (50.0) | 3 (37.5) |
| 23 | 23 (69.7) | 6 (75.0) | 2 (66.7) | 2 (100.0) | 1 (100.0) |
| 24 | 19 (100.0) | 10 (100.0) | 5 (83.3) | 7 (100.0) | 4 (100.0) |
| 25 | 7 (58.3) | 6 (85.7) | 4 (80.0) | 3 (100.0) | 3 (100.0) |
| 26† | 8 (80.0) | 6 (100.0) | 7 (100.0) | - | - |
| 27† | 5 (71.4) | 5 (55.6) | 3 (100) | 1 (100.0) | - |
| **Total** | 760 | 697 | 554 | 488 | 320 |
| **Mean (SD)** | 81.1% (0.11) | 83.7% (0.14) | 77.9% (0.18) | 82.2% (0.18) | 86.5% (0.17) |
| **Median (IQR)** | 82.8% (79–87) | 86.0% (81–93) | 83.3% (75–90) | 90.5% (78–100) | 90.9% (75–100) |

† During 2019 and 2020, the two centers (centers 26 and 27) were closed and no longer functioning as HIV treatment centers.

center with the highest gMTR use (91%) were 32.5% lower than those in treatment centers with no gMTR use at all.

## Discussion

Our study investigated compliance to Dutch HIV treatment guidelines, which closely follow the DHHS guidelines, and the choice of antiretroviral therapy among people with HIV initiating first-line ART in the Netherlands. Our results showed that the overall guideline compliance was more than 80%, with minor variability over the years and between HIV treatment centers. After the introduction of generic NRTIs backbones in the Netherlands at the end of 2017, the average monthly medication cost per patient in 2020 decreased by 22.9%. This decrease was even larger (up to 30%) in centers that chose to initiate treatment-naïve people with HIV on a gMTR.

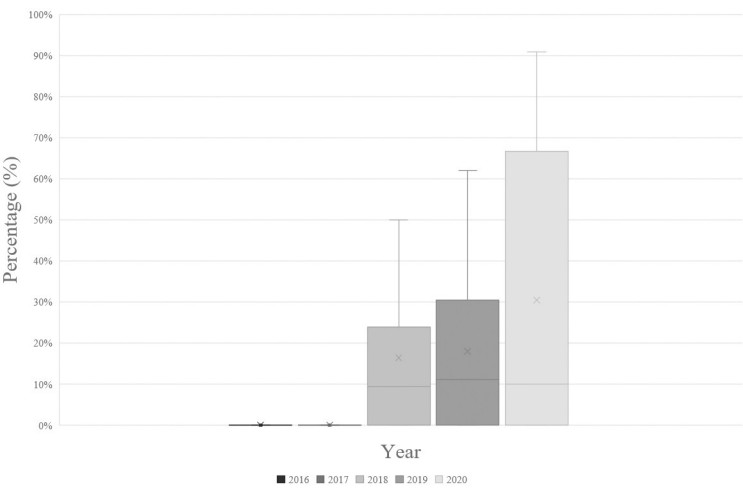

**Fig 1. Box plot with the distribution of % generic MTR across all treatment centers at a national level from 2016 to 2020.**

| | 1 | 2 | 3 | 4 | 5 | 6 | 7 | 8 | 9 | 10 | 11 | 12 | 13 | 14 | 15 | 16 | 17 | 18 | 19 | 20 | 21 | 22 | 23 | 24 | 25 | 26 | 27 |
|---|---|---|---|---|---|---|---|---|---|---|---|---|---|---|---|---|---|---|---|---|---|---|---|---|---|---|---|
| 2018 | 16% | 35% | 11% | 9% | 0% | 0% | 3% | 29% | 8% | 24% | 4% | 18% | 92% | 0% | 8% | 0% | 0% | 18% | 22% | 0% | 8% | 13% | 0% | 40% | 50% | 0% | 33% |
| 2019 | 62% | 53% | 3% | 15% | 0% | 11% | 17% | 30% | 4% | 25% | 0% | 0% | 94% | 0% | 0% | 0% | 38% | 0% | 25% | 13% | 0% | 33% | 0% | 29% | 33% | 0% | 0% |
| 2020 | 91% | 54% | 5% | 67% | 8% | 0% | 5% | 78% | 0% | 60% | 0% | 25% | 77% | 11% | 0% | 0% | 0% | 10% | 67% | 89% | 0% | 67% | 0% | 75% | 33% | 0% | 0% |

**Fig 2. Distribution of generic MTR usage across treatment centers from 2018–2020, percentage of generic MTR prescriptions per treatment center per year.**

**Table 5. Annual costs and distribution of the DHHS-recommended initial regimens from 2016–2020.**

| Regimen | Year 2016 € | 2017 € | 2018 € | 2019 € | 2020 |
|---|---|---|---|---|---|
| **Single-tablet regimens (€, %)** | | | | | |
| BIC/TAF/FTC | | | 48,546 (10.0) | 276,892 (68.2) | 154,512 (62.1) |
| DTG/ABC/3TC | 317,347 (44) | 250,821 (38) | 157,325 (32) | 39,556 (9.7) | 14,179 (5.7) |
| EVG/c/TAF/FTC | 172,227 (23.9) | 218,277 (33.0) | 159,333 (32.8) | | |
| EVG/c/TDF/FTC | 75,621 (10.5) | 43,628 (8.9) | 13,573 (2.8) | | |
| **Branded multi-tablet regimens (€, %)** | | | | | |
| DTG, TDF/FTC | 107,405 (14.9) | 78,276 (11.8) | | | |
| RAL, TDF/FTC | 3,518 (0.5) | 4,414 (0.74) | | | |
| DTG, TAF/FTC | | 58,621 (8.9) | 43,423 (8.9) | 14,060 (3.5) | 5,344 (2.1) |
| RAL, TAF/FTC | | 6,612 (1.0) | 8,769 (1.8) | 1,090 (0.3) | 1,083 (0.4) |
| **Generic multi-tablet regimens† (€, %)** | | | | | |
| DTG, TDF/FTC | | | 47,626 (9.8) | 70,218 (17.3) | 72,578 (29.2) |
| RAL, TDF/FTC | | | 7,454 (1.5) | 4,336 (1.1) | 1,249 (0.5) |
| **Other regimens (€, %)** | | | | | |
| DRV/RTV, TDF/FTC | 43,821 (6.1) | | | | |
| **Total** | 719,939 | 660,648 | 486,049 | 406,152 | 248,944 |

† Calculated as antiretroviral therapy containing a generic backbone (TDF/FTC), introduction in 2017

3TC: lamivudine, ABC: abacavir, BIC: bictegravir, DRV: darunavir, DTG: dolutegravir, EVG/c: elvitegravir/cobicistat, FTC: emtricitabine, RAL: raltegravir, RTV: ritonavir, TAF: tenofovir alafenamide, TDF: tenofovir disoproxil fumarate

In the Netherlands, people with HIV receive care in specialized HIV treatment centers across the country, where approximately 100 physicians are responsible for initiating ART [15]. At each center, at least two infectious disease physicians and at least two specialized HIV nurses who are responsible for the care of people with HIV are employed, with support from medical microbiologists, pharmacists, and other medical specialists. Almost all HIV physicians are members of the NVHB who have issued treatment guidelines for starting first-line ART based on DHHS guidelines [1].

The high compliance to treatment guidelines in almost all treatment centers can be explained by multiple factors. First, Dutch HIV physicians are familiar with these guidelines, which are easy to follow and are of practical use. Furthermore, SHM publishes a yearly report (SHM monitoring report) that includes data on ART use at the national level, which can be used as a reference [17]. Finally, during these years, each treatment center was visited every three years for quality and safety management systems through the Harmonization of Quality Assessment in the Healthcare Sector (in Dutch, Harmonisatie Kwaliteitsbeoordeling in de Zorgsector, HKZ 158). These standards were developed by the NVHB in the direction of the HKZ as field parties, with the subsequent approval of the NVHB members [18]. Prescribing ART within or outside of the guidelines was included in this assessment.

Although high compliance to treatment guidelines is usually interpreted as a positive out-come, physicians will always be confronted with individual patient cases that do not meet the general guideline criteria for starting ART. Therefore, 100% guideline compliance was difficult to obtain. One of the clearest situations for people with a new HIV diagnosis could be pregnancy, for which more specific treatment recommendations have been formulated, which deviate from the non-pregnant situation (see the section in the DHHS guidelines on antiretroviral regimen considerations for initial therapy based on specific clinical scenarios [3]). In the

**Table 6. Average monthly price distribution of DHHS-recommended initial regimens per person per treatment center from 2016–2020.**

| Center | Year 2016 ART (€) | 2017 ART (€) | 2018 ART (€) | 2019 ART (€) | 2020 ART (€) | Δ *2016–2020* ART (€) |
|---|---|---|---|---|---|---|
| 1 | 927.58 | 920.79 | 876.98 | 722.84 | 634.11 | -293.47 |
| 2 | 953.44 | 950.57 | 798.50 | 753.76 | 750.34 | -203.10 |
| 3 | 967.35 | 953.91 | 913.26 | 890.53 | 859.80 | -107.55 |
| 4 | 965.90 | 948.07 | 898.00 | 862.21 | 697.58 | -268.32 |
| 5 | 928.33 | 919.01 | 926.28 | 899.00 | 852.72 | -75.62 |
| 6 | 919.28 | 940.89 | 914.86 | 867.28 | 873.69 | -45.60 |
| 7 | 920.76 | 969.74 | 953.43 | 849.23 | 859.80 | -60.97 |
| 8 | 954.41 | 970.27 | 825.46 | 819.16 | 671.30 | -283.12 |
| 9 | 979.20 | 987.56 | 903.83 | 894.76 | 872.95 | -106.25 |
| 10 | 964.05 | 984.60 | 876.80 | 828.00 | 720.52 | -243.53 |
| 11 | 937.37 | 931.94 | 902.72 | 908.11 | 872.95 | -64.42 |
| 12 | 923.04 | 945.82 | 868.75 | 899.00 | 807.19 | -115.85 |
| 13 | 1,036.88 | 969.48 | 650.61 | 640.57 | 670.60 | -366.27 |
| 14 | 903.65 | 918.43 | 917.74 | 908.13 | 843.72 | -59.92 |
| 15 | 941.84 | 938.14 | 954.50 | 944.63 | 948.17 | 6.32 |
| 16 | 909.57 | 950.33 | 917.33 | 899.00 | 872.95 | -36.62 |
| 17 | 959.76 | 938.55 | 929.08 | 790.85 | 872.95 | -86.81 |
| 18 | 952.03 | 968.86 | 875.85 | 899.00 | 846.65 | -105.38 |
| 19 | 911.97 | 913.67 | 849.58 | 826.90 | 697.58 | -214.39 |
| 20 | 942.72 | 955.45 | 906.33 | 862.95 | 639.13 | -303.59 |
| 21 | 961.37 | 929.08 | 886.09 | 899.00 | 872.95 | -88.42 |
| 22 | 937.98 | 928.11 | 872.52 | 863.70 | 697.58 | -240.39 |
| 23 | 933.43 | 910.00 | 910.00 | 899.00 | 872.95 | -60.48 |
| 24 | 910.06 | 949.11 | 827.46 | 816.60 | 675.66 | -234.39 |
| 25 | 1,014.41 | 1,086.63 | 772.42 | 802.86 | 790.05 | -224.36 |
| 26† | 904.88 | 917.33 | 908.43 | - | - | - |
| 27† | 934.13 | 954.23 | 821.05 | 899.00 | - | - |
| **Mean (SD)** | 944.27 (31.61) | 950.02 | 876.22 | 820.22 | 732.37 | |

† During 2019 and 2020, the two centers (centers 26 and 27) were closed and no longer functioning as HIV treatment centers.

Netherlands, approximately 5% of patients initiating ART do so during pregnancy, and this has often been ritonavir-boosted darunavir, which has been removed from the DHHS guidelines for non-pregnant people with HIV since 2017 [15]. Furthermore, several treatment centers participated in an acute HIV study protocol mandating the initiation of quadruple ART [19]. Because of these and other more specific patient-related conditions, 100% compliance to the DHHS "recommended initial regimens for most people with HIV" guidelines is difficult to obtain but also should not be our goal.

Our data demonstrated an uptake of gMTR at a national level of almost 40% of all new people with HIV initiating ART at the end of our follow-up period in 2020. Our study provides three important observations regarding gMTR uptake in the Netherlands.

First, for a long time, physicians and patients preferred STR to start treatment because of its low pill burden and higher convenience. STRs are widely regarded as the gold standard for ART because of their ability to ensure maximum adherence and efficacy and are often actively promoted for this reason. Various studies have consistently shown that MTRs can be as

effective as STRs, especially when the number of intake moments is similar, typically involving once-daily ART [20]. The perception that MTR is associated with a higher pill burden and possibly reduced adherence and effectiveness was not supported in a number of studies, including our own research [21–25]. All of these studies failed to find any significant differences in adherence between MTRs and STRs.

Second, while Dutch HIV physicians appear to comply to treatment guidelines almost unanimously, a large variability in the uptake of gMTR can be detected between various treatment centers. For instance, while eight treatment centers did not utilize gMTRs at all, five treatment centers prescribed gMTRs to over 75% of their patients with HIV initiating ART in 2020. This notable diversity in prescription practices may be indicative of differing beliefs, particularly regarding the use of generic medications among HIV physicians, as evidenced by a review and two separate studies conducted in France [26–29]. Concerns were raised regarding the supply chain, fixed-dose combinations, adherence, dosing frequency, and pill burden. Such (lack of) belief in generics or the anticipated adherence benefit of an STR greatly influences the decision to start a gMTR or a branded STR in treatment-naïve people with HIV. Another explanation for the variability between treatment centers in the utilization of gMTRs may be differences in the patient population [30]. For instance, when confronted with a newly diagnosed person with HIV and a language barrier, STR initiation may be a better choice to promote adherence and limit medication mistakes, thus improving virological efficacy. Individuals with HIV and complex needs may be more common in certain treatment centers, thus explaining the higher proportion of STRs at these sites.

Third, regarding the decision of HIV physicians and people with HIV to initiate gMTR, there was a notable reduction in medication costs. The average monthly medication cost per patient decreased from €950 to €732 (-22.9%) between 2017 and 2020. Treatment centers with the highest uptake of gMTR prescriptions experienced an even more significant impact, with medication costs lowered by up to almost 30%. Part of this reduction in costs could also be attributed to a reduction in the price of STRs in the Netherlands, although this was only a small effect (<2%; Table 1).

Some physicians may not realize the major cost impact of choosing gMTR over STR. However, even if they do, they may hesitate to mention the cost to newly diagnosed people. Nevertheless, contrary to common assumptions, several studies have provided substantial evidence showing that a majority of people with HIV, including both experienced and newly diagnosed individuals, are receptive to the idea of switching between STR and MTR, with acceptance rates often hovering around 50% [21, 23–25]. These findings highlight the potential of discussing the price of antiretroviral therapy (ART) as a viable approach in patient-centered care [31]. We suggest that it would be useful to prioritize the cheapest ART at the top of the DHHS guidelines, or if this is not possible because of differences in healthcare systems between the US and the Netherlands, that the NVHB modifies the DHHS table.

Our study has several limitations. Most importantly, we did not collect data on the persistence of gMTR or STR in the SHM cohort. It is possible that patients switch from gMTR to STR after starting ART, so the benefit of lower drug costs may not last longer period of time. An alternative policy for initiating ART that some doctors or treatment centers may follow is starting an STR and switching to a gMTR a few months later when good adherence and an initial virological response are demonstrated. We did not have information on the outcomes of such an approach in our current dataset, as we only investigated the starting ART regimen during the first month. In the aforementioned STR-to-gMTR switches, the benefit of lower medication costs is only present at a later stage.

Our primary interest in this study was to gain insights into broader patterns at the national and treatment center levels rather than focusing on individual-level analysis. Therefore, our

dataset was deliberately limited in scope, excluding specific patient characteristics, such as age, sex, comorbidities, co-medication, and ethnicity, which could potentially influence the choice of ART on an individual basis. By focusing on the national and treatment center levels, we aimed to understand the overall trends and variations in ART prescriptions, providing valuable information for public health strategies and treatment center management.

Furthermore, our data did not provide information on the viral load, adherence, or other clinical outcomes. Although real-world data might not have the same level of control as clinical trial data, numerous studies have demonstrated that real-world outcomes for ART are generally comparable to those reported in clinical trials. Specifically, these studies have shown that there is no significant difference in outcomes between MTR and STR in terms of viral suppression and adherence [20, 24, 32, 33]. Despite TAF's better clinical profile regarding adverse events compared to TDF, our data showed a greater use of TDF in this population. This preference for TDF could be attributed to its earlier availability and lower cost, which may influence prescribing practices. However, the long-term economic impact of this choice should be considered, as the adverse events associated with TDF could potentially increase the overall HIV treatment burden over time.It is important to acknowledge the limitations of our dataset and recognize that future research should investigate the clinical outcomes of patients receiving different first-line ART regimens in real-world settings to further validate these findings.

Finally, another limitation was that we analyzed the prescription of ART regimens rather than actual dispensing in the pharmacy. Although theoretically possible, it is unlikely that branded formulations were prescribed because the policy in the Netherlands always prescribes generic formulations. We did not analyze the actual costs of the dispensed ART regimens, but this could be a topic for further research based on data from Dutch pharmacies provided by the Foundation for Pharmaceutical Statistics (SFK).

## Conclusions

Our study provides insights into compliance to the DHHS guidelines for initial ART regimens, associated costs, and variability between treatment centers in the Netherlands. Our findings highlight the need for efforts to promote cost-effective prescribing practices across all treatment centers. Future studies should explore the reasons for the observed variability and the impact of different initial regimens on clinical outcomes and overall health care costs.

## Supporting information

**S1 Table. Distribution of STR versus branded MTR versus generic MTR from 2016 to 2020.**
(DOCX)

## Acknowledgments

We thank Peter Reiss for his contribution to the conceptual process of data collection and analysis.

## Author Contributions

**Conceptualization:** Piter Oosterhof, Ferdinand W. N. M. Wit, Matthijs van Luin, Marc van der Valk, Kees Brinkman, David M. Burger.

**Data curation:** Piter Oosterhof, Ferdinand W. N. M. Wit.

**Formal analysis:** Piter Oosterhof.

**Methodology:** Piter Oosterhof.

**Supervision:** David M. Burger.

**Writing – original draft:** Piter Oosterhof, David M. Burger.

**Writing – review & editing:** Ferdinand W. N. M. Wit, Matthijs van Luin, Marc van der Valk, Kees Brinkman.

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
