## [Decision Letter · Decision Letter 0]

25 Apr 2024

PONE-D-24-10391First-line antiretroviral therapy initiation for newly diagnosed people with HIV in the Netherlands: a retrospective analysis from 2016 to 2020.PLOS ONE

Dear Dr. Oosterhof,

Thank you for submitting your manuscript to PLOS ONE. After careful consideration, we feel that it has merit but does not fully meet PLOS ONE’s publication criteria as it currently stands. Therefore, we invite you to submit a revised version of the manuscript that addresses the points raised during the review process.

**ACADEMIC EDITOR: **

**MAJOR COMMENTS**

*Table 4. Adherence rate * Here and so on in many other cases, the term adherence might be misunderstood in the manuscript. According to WHO, adherence “is the extent to which a person's behavior – taking medication, attending scheduled clinic appointments, following a diet and/or changing lifestyle – corresponds with care and treatment plans”. Compliance rate or compliance with Dutch guidelines could be more suitable. On the other hand, adherence is most appropriately used in Discussion (lines 258, 262, 263).

In line with the third observation done by authors in Discussion (lines 282-287), I missed a detailed economic impact of the generic drugs. This calculation could be done from percentages known (and depicted in Table 3). How much decline can be attributed to the availability of generics? How much can be attributed to other factors in addition to the indicated small reduction in the price of STRs? For instance, some regimens (BIC/TAF/FTC, EVG/c/TAF/FTC, DRV/c+TAF/FTC,…) contribute unequally throughout the period. I recommend perform this calculation, which would improve the conclusions obtained.

Figure 3 (cited in line 205) is missing.

**MINOR COMMENTS**

Line 165. Three of the 27 treatment centers  Better: Three*
out of
*the 27…Centers firstly listed in Table 4 must fully be depicted as supplementary material. The results section includes some statements discussing the findings (from line 129 to 133; from line 151 to 154). They should be allocated in Discussion section.*Table 3. Percentage of patients on DHHS-recommended initial regimens from 2016 – 2020 * Percentages are shown only for each regimen but they must be given for all the options within them. In addition, the heading of this table may simply be “Patients on DHHS-recommended…”. ==============================

We look forward to receiving your revised manuscript.

Kind regards,

Carmen María González-Domenech, Ph.D.

Academic Editor

PLOS ONE

“PO received personal consulting fees from ViiV Healthcare, Gilead, Merck Sharp, and Dohme. FW has been served on scientific advisory boards for ViiV Healthcare and Gilead Sciences. MvdV has served on advisory boards for Gilead Sciences (Merck, Sharp, & Dohme) and ViiV Healthcare. MvdV received independent scientific grant support from Gilead Sciences, Merck, Sharp, & Dohme, and ViiV Healthcare, all of which were paid to his institution (Amsterdam University Medical Centers, Amsterdam, Netherlands). KB received personal consultation fees from Merck, Gilead, and ViiV. KB has received a writing honorarium from UpToDate. The remaining authors declare that they have no conflicts of interest.”

3. For studies involving third-party data, we encourage authors to share any data specific to their analyses that they can legally distribute. PLOS recognizes, however, that authors may be using third-party data they do not have the rights to share. When third-party data cannot be publicly shared, authors must provide all information necessary for interested researchers to apply to gain access to the data. (https://journals.plos.org/plosone/s/data-availability#loc-acceptable-data-access-restrictions)

a) A description of the data set and the third-party source

b) If applicable, verification of permission to use the data set

c) Confirmation of whether the authors received any special privileges in accessing the data that other researchers would not have

d) All necessary contact information others would need to apply to gain access to the data

Additional Editor Comments:

This manuscript retrospectively analyzed the antiretroviral therapy of initiation for naïve HIV patients diagnosed between January 2016 and December 2020. Type and agreement with guidelines by Dutch Department of Health and Human Services were assessed as well as monthly cost.

It is a very interesting overview of that topic in the Netherlands. The manuscript is well-written but I have some major and other minor comments from myself and the reviewers, which should be addressed as much as possible before publication.

MAJOR COMMENTS

Table 4. Adherence rate Here and so on in many other cases, the term adherence might be misunderstood in the manuscript. According to WHO, adherence “is the extent to which a person's behavior – taking medication, attending scheduled clinic appointments, following a diet and/or changing lifestyle – corresponds with care and treatment plans”. Compliance rate or compliance with Dutch guidelines could be more suitable. On the other hand, adherence is most appropriately used in Discussion (lines 258, 262, 263).

In line with the third observation done by authors in Discussion (lines 282-287), I missed a detailed economic impact of the generic drugs. This calculation could be done from percentages known (and depicted in Table 3). How much decline can be attributed to the availability of generics? How much can be attributed to other factors in addition to the indicated small reduction in the price of STRs? For instance, some regimens (BIC/TAF/FTC, EVG/c/TAF/FTC, DRV/c+TAF/FTC,…) contribute unequally throughout the period.

Figure 3 (cited in line 205) is missing.

MINOR COMMENTS

165 period. Three of the 27 treatment centers Three out of the 27…

Centers firstly listed in Table 4 must fully be depicted as supplementary material.

The results section includes some statements discussing the findings (from line 129 to 133; from line 151 to 154). They should be allocated in Discussion section.

Table 3. Percentage of patients on DHHS-recommended initial regimens from 2016 – 2020 Percentages are shown only for each regimen but they must be given for all the options within them. In addition, the heading of this table may simply be “Patients on DHHS-recommended…”.

Reviewers' comments:

Reviewer's Responses to Questions

**Comments to the Author**

1. Is the manuscript technically sound, and do the data support the conclusions?

Reviewer #1: Yes

Reviewer #2: Yes

Reviewer #3: Yes

2. Has the statistical analysis been performed appropriately and rigorously? 

Reviewer #1: Yes

Reviewer #2: Yes

Reviewer #3: Yes

3. Have the authors made all data underlying the findings in their manuscript fully available?

Reviewer #1: Yes

Reviewer #2: Yes

Reviewer #3: Yes

4. Is the manuscript presented in an intelligible fashion and written in standard English?

Reviewer #1: Yes

Reviewer #2: Yes

Reviewer #3: Yes

5. Review Comments to the Author

Reviewer #1: 1. 114-123: Overuse of passive voice in this paragraph, should be changed to active voice to be better understood.

2. Discussion: I miss the explanation of why there is a greater use of TDF vs TAF in this population in spite of having, the second one, a better clinical profile of adverse events. This could affect the prize of HIV burden in long term. Try to explain this in few words.

3. Bibliography mistakes:

- 417: avoid (London, England).

- 422: avoid (London, England).

- 435: avoid dots.

- 437: 2021. 2021.

- 438: c2023.

- 440: c2023.

- 442: Netherlands2022

- 452: avoid (London, England).

- 480: BMJ Open.

Reviewer #2: Oosterhof et al analyze the initial ART regimens in the Netherlands in a large cohort of 3,445 HIV naïve patients starting ART between January 2016 and December 2020.They observed an increasing tendence to prescribe integrase inhibitor containing regimens from 77.3% in 2016 to 87.8% in 2020. Most patients received STR regimens although with a decreasing rate from 81.3% in 2016 to 60.3% in 2020. The gMTRs showed a steady increase from 17.8% in 2018 to 37.8% in 2020. This fact implied a cost reduction of the first-line ART regimen per patient from 17.8% in 2018 to 37.8% in 2020. The adherence rate to DHHS-recommended initial regimens increased from 82.8% in 2016 to 90.0% in 2020, not influenced by the gMTRs prescription increase.

Major points

1.This work reinforces previous studies showing that ART adherence rate is not always linked to STR and this is a key point of the manuscript. However all the enrolled patients received their ART once-daily (QD regimens) and this is important to emphasize in the discussion. The number of intake moments is important with perfect adherence linked to once-daily ART.

2.The fact that the STR and gMTR prescription data provided here represents the situation at the start of HIV therapy and not on the persistence of STR and gMTR is the main pitfall of the study. Could you provide some data on the number of ART switch per patient or study center and analyzed year ?

Minor points

1. I do not see why 100% adherence to DHHS recommended regimens is not desirable. (pages 17, line 241,and 18, line 244.250). Perhaps this sentence could be changed to “ difficult to obtain” due to pregnancy-due specific ART regimens and some study protocols that pushed the ART from the DHHS recommendations.

2. “Finally , another limitation was that we analyzed the prescription of ART regimens rather than specific formulation” (Page 21, lines 326-327).Do you mean single antiretrovirals and not combined SRTs and gMTRs regimens when to mention specific formulations ?

Reviewer #3: The authors present a retrospective study on first-line antiretroviral therapy for people newly diagnosed with HIV in the Netherlands. The article is well written and provides data of interest.

Two minor consdierations

- Please consider improving Figure 2. It is complex to interpret

- Page 16, line 205: costs per person per month for initiating ART (Figure 3). I cannot find this figure

6. PLOS authors have the option to publish the peer review history of their article (what does this mean?). If published, this will include your full peer review and any attached files.

Reviewer #1: No

Reviewer #2: No

Reviewer #3: No

---

## [Author Response · Author response to Decision Letter 0]

8 Jul 2024

Response to reviewers’ comments

ACADEMIC EDITOR:

MAJOR COMMENTS

1. Table 4. Adherence rate Here and so on in many other cases, the term adherence might be misunderstood in the manuscript. According to WHO, adherence “is the extent to which a person's behavior – taking medication, attending scheduled clinic appointments, following a diet and/or changing lifestyle – corresponds with care and treatment plans”. Compliance rate or compliance with Dutch guidelines could be more suitable. On the other hand, adherence is most appropriately used in Discussion (lines 258, 262, 263).

We understand the importance of terminology precision and agree that "compliance" more accurately describes the adherence to Dutch guidelines in the context of our study. We will replace "adherence rate" with "compliance rate" throughout the manuscript where it refers to guideline adherence. However, we will retain "adherence" in the Discussion section where it pertains to patient behavior, aligning with WHO definitions. Below are specific actions we will take in response to this comment:

I. Replace "adherence rate" with "compliance rate" in Table 4 and other relevant sections of the manuscript.

II. Ensure consistent use of "adherence" in the Discussion section as per WHO definitions.

We believe these changes will improve clarity and accuracy in our manuscript.

2. In line with the third observation done by authors in Discussion (lines 282-287), I missed a detailed economic impact of the generic drugs. This calculation could be done from percentages known (and depicted in Table 3). How much decline can be attributed to the availability of generics? How much can be attributed to other factors in addition to the indicated small reduction in the price of STRs? For instance, some regimens (BIC/TAF/FTC, EVG/c/TAF/FTC, DRV/c+TAF/FTC,…) contribute unequally throughout the period. I recommend perform this calculation, which would improve the conclusions obtained.

Regarding the economic impact analysis of the generic drugs in our manuscript. In response to your suggestion, we have performed a detailed calculation and included the results in the manuscript. Specifically, we have addressed the economic impact of different ART regimens over the years 2016 to 2019 and analyzed how much of the cost decline can be attributed to the availability of generics and other factors.

We have included the following in the revised manuscript:

1) Annual costs and distribution of the DHHS-recommended initial regimens from 2016 - 2020: The detailed breakdown of the costs associated with different ART regimens from 2016 to 2019 is now presented in Table 5, illustrating changes in the economic landscape of ART prescriptions.

2) Text in the results section: “The total expenditure on ART regimens decreased from €719,938 in 2016 to €248,944 in 2020. In the last two years, the regimens BIC/TAF/FTC and generic DTG, TDF/FTC had the largest share of the costs”

We believe these additions provide a more comprehensive understanding of the economic impact of the availability of generic drugs and enhance the conclusions of our study.

Figure 3 (cited in line 205) is missing.

We apologize for the oversight. The reference to Figure 3 in line 205 was incorrect and should not have been included. Due to the removal of the old Figure 2, the figures have been renumbered. The figure intended to be referenced as Figure 3 is not correctly labeled as Figure 3 in the current manuscript. We have adjusted the figure citations throughout the text to reflect the correct numbering.

MINOR COMMENTS

1. Line 165. Three of the 27 treatment centers Better: Three out of the 27…

We agree that "Three out of the 27" is a clearer and more precise phrasing. We will make this change in the manuscript.

2. Centers firstly listed in Table 4 must fully be depicted as supplementary material. 

Addendum information by email sent on 17-06-2024 regarding the above comment:

Thank you for reaching to me.

My comment is referred to the name and address of the centers listed in Table 4. These centers must be identified not only by a number. However, this information could be added as supplementary material.

Hence, in addition to table 4, another showing specific identifying data from centers

The ethical guidelines and privacy regulations required that the treatment centers be coded to prevent their identification by the researchers. This was necessary to ensure the privacy of the involved centers. Our manuscript's ethics section explains this.

“ Since our study involved analyzing pre-existing data and the data collection process aligns with the routine operations of the SHM, we deemed it exempt from additional ethical approval requirements. We coded the treatment centers to ensure investigators could not identify them. Additionally, we pseudonymized data from each treatment center to protect privacy and confidentiality while maintaining traceability for analysis per calendar year.”

In summary, it was mandatory that the centers be coded so that they could not be identified by the researchers to ensure privacy. Given the ethical and privacy considerations, it is not possible to include the specific names and addresses of the treatment centers in the main table.

3. The results section includes some statements discussing the findings (from line 129 to 133; from line 151 to 154). They should be allocated in Discussion section.

Thank you for pointing out this issue. We understand the importance of maintaining a clear distinction between results and discussion. We have revised the statements (in the mentioned lines) in the Results section to ensure they present factual observations without interpretation. 

4. Table 3. Percentage of patients on DHHS-recommended initial regimens from 2016 – 2020 Percentages are shown only for each regimen but they must be given for all the options within them. In addition, the heading of this table may simply be “Patients on DHHS-recommended…”.

Thank you for your suggestion. We have revised Table 3 to include percentages for all options within each regimen, and simplify the table heading as recommended.

Reviewer: #1

1. 114-123: Overuse of passive voice in this paragraph, should be changed to active voice to be better understood.

Thank you for your feedback regarding the use of passive voice in the Ethics section. We have revised the paragraph (mentioned lines) to use active voice for better clarity and understanding.

2. Discussion: I miss the explanation of why there is a greater use of TDF vs TAF in this population in spite of having, the second one, a better clinical profile of adverse events. This could affect the prize of HIV burden in long term. Try to explain this in few words.

In response to the feedback regarding the explanation for the greater use of TDF compared to TAF, we have added the following text to our discussion section:

"Despite TAF's better clinical profile regarding adverse events compared to TDF, our data showed a greater use of TDF in this population. This preference for TDF could be attributed to its earlier availability and lower cost, which may influence prescribing practices. However, the long-term economic impact of this choice should be considered, as the adverse events associated with TDF could potentially increase the overall HIV treatment burden over time."

We believe this addition addresses the concern by providing a concise explanation for the observed preference for TDF over TAF, considering both clinical and economic factors.

3. Bibliography mistakes:

- 417: avoid (London, England).

- 422: avoid (London, England).

- 435: avoid dots.

- 437: 2021. 2021.

- 438: c2023.

- 440: c2023.

- 442: Netherlands2022

- 452: avoid (London, England).

- 480: BMJ Open.

We have carefully reviewed the identified issues and made the necessary corrections. We appreciate your detailed comments.

Reviewer: #2

Oosterhof et al analyze the initial ART regimens in the Netherlands in a large cohort of 3,445 HIV naïve patients starting ART between January 2016 and December 2020.They observed an increasing tendence to prescribe integrase inhibitor containing regimens from 77.3% in 2016 to 87.8% in 2020. Most patients received STR regimens although with a decreasing rate from 81.3% in 2016 to 60.3% in 2020. The gMTRs showed a steady increase from 17.8% in 2018 to 37.8% in 2020. This fact implied a cost reduction of the first-line ART regimen per patient from 17.8% in 2018 to 37.8% in 2020. The adherence rate to DHHS-recommended initial regimens increased from 82.8% in 2016 to 90.0% in 2020, not influenced by the gMTRs prescription increase.

Major points

1.This work reinforces previous studies showing that ART adherence rate is not always linked to STR and this is a key point of the manuscript. However all the enrolled patients received their ART once-daily (QD regimens) and this is important to emphasize in the discussion. The number of intake moments is important with perfect adherence linked to once-daily ART.

Thank you for the valuable comment. Due to our misplaced use of the word "adherence," where I meant "compliance" to the guidelines, the reviewer may have misunderstood the context. This valuable feedback highlights that adjusting "adherence" to "compliance" will make my manuscript clearer.

Additionally, we appreciate the emphasis on the importance of once-daily (QD) regimens. All enrolled patients received their ART as once-daily regimens, which is an aspect to mention in the discussion (line 269). The number of intake moments is indeed important, with good adherence often linked to once-daily ART. We will update the manuscript accordingly to reflect this point more clearly (lines 277-279).

2.The fact that the STR and gMTR prescription data provided here represents the situation at the start of HIV therapy and not on the persistence of STR and gMTR is the main pitfall of the study. Could you provide some data on the number of ART switch per patient or study center and analyzed year ?

We acknowledge the fact that the STR and gMTR prescription data provided here represents the situation at the start of HIV therapy and not on the persistence of STR and gMTR is a limitation of our study. As mentioned in the discussion section, "Our study has several limitations. Most importantly, we did not collect data on the persistence of gMTR or STR in the SHM cohort."

Unfortunately, our current dataset does not include detailed information on ART switches per patient or per study center across the analyzed years. However, we recognize the importance of this data and how it could enhance the understanding of treatment persistence and switching patterns. It is important to note that the SHM already reports annually on the persistence of various therapies. These reports provide valuable insights into long-term treatment patterns and outcomes.

Minor points

1. I do not see why 100% adherence to DHHS recommended regimens is not desirable. (pages 17, line 241,and 18, line 244.250). Perhaps this sentence could be changed to “ difficult to obtain” due to pregnancy-due specific ART regimens and some study protocols that pushed the ART from the DHHS recommendations.

We appreciate your feedback regarding the desirability of 100% adherence to DHHS recommended regimens. We have revised the sentence to better reflect the practical challenges in achieving this level of compliance due to specific patient-related conditions, such as pregnancy-specific ART regimens and study protocols that necessitate deviations from the DHHS recommendations.

The revised sentence now reads: "Because of these and other more specific patient-related conditions, 100% compliance with the DHHS ‘recommended initial regimens for most people with HIV’ guidelines is difficult to obtain but also should not be our goal." This adjustment emphasizes that achieving 100% compliance is challenging and acknowledges that it may not be realistic or necessary in all clinical scenarios. We believe this change addresses your concern and provides a clearer rationale for the observed variability in adherence to the guidelines.

2. “Finally , another limitation was that we analyzed the prescription of ART regimens rather than specific formulation” (Page 21, lines 326-327).Do you mean single antiretrovirals and not combined SRTs and gMTRs regimens when to mention specific formulations ?

Clarification is needed in our statement about the analysis of ART regimens. To clarify, we intended to communicate that our analysis focused on the prescriptions of ART regimens rather than the actual dispensing of medications at the pharmacy. Therefore, we have revised the sentence to better reflect this distinction.

The revised sentence now reads: "Finally, another limitation was that we analyzed the prescription of ART regimens rather than the actual dispensing in the pharmacy." This adjustment clarifies that our study examined the prescribed treatments and not the specific formulations dispensed by the pharmacy.

Reviewer: #3

The authors present a retrospective study on first-line antiretroviral therapy for people newly diagnosed with HIV in the Netherlands. The article is well written and provides data of interest.

Two minor considerations

1. Please consider improving Figure 2. It is complex to interpret

Thank you for your feedback regarding Figure 2. We understand that the figure was complex to interpret, and we have made several improvements to enhance its clarity and readability. We believe these adjustments will make Figure 2 more accessible and easier to interpret for readers. We appreciate your valuable feedback, which has helped us enhance the quality of our manuscript.

2. Page 16, line 205: costs per person per month for initiating ART (Figure 3). I cannot find this figure

The reference to Figure 3 in line 205 was incorrect and should not have been included. Due to the removal of the old Figure 2, the figures have been renumbered. The figure intended to be referenced as Figure 3 is not correctly labeled as Figure 3 in the current manuscript. We have adjusted the figure citations throughout the text to reflect the correct numbering.

Competing Interests section:

“PO received personal consulting fees from ViiV Healthcare, Gilead, Merck Sharp, and Dohme. FW has been served on scientific advisory boards for ViiV Healthcare and Gilead Sciences. MvdV has served on advisory boards for Gilead Sciences (Merck, Sharp, & Dohme) and ViiV Healthcare. MvdV received independent scientific grant support from Gilead Sciences, Merck, Sharp, & Dohme, and ViiV Healthcare, all of which were paid to his institution (Amsterdam University Medical Centers, Amsterdam, Netherlands). KB received personal consultation fees from Merck, Gilead, and ViiV. KB has received a writing honorarium from UpToDate. The remaining authors declare that they have no conflicts of interest.”

I confirm that this not alter my adherence to all PLOS ONE policies. And thereby:

This does not alter our adherence to PLOS ONE policies on sharing data and materials.

---

## [Editor Report · Decision Letter 1]

16 Jul 2024

First-line antiretroviral therapy initiation for newly diagnosed people with HIV in the Netherlands: a retrospective analysis from 2016 to 2020.

PONE-D-24-10391R1

Dear Dr. Piter Oosterhof, PharmD,

We’re pleased to inform you that your manuscript has been judged scientifically suitable for publication and will be formally accepted for publication once it meets all outstanding technical requirements.

Kind regards,

Carmen María González-Domenech, Ph.D.

Academic Editor

PLOS ONE

Additional Editor Comments (optional):

Thank you for properly addressing all the issues pointed by the reviewers and the academic editor. In addition, the minor mistakes noticed have also been amended. Thus, your manuscript is ready to be publised by PLoS One journal in the current version. Thank you for submitting your valuable work to our journal.

---

## [Editor Report · Acceptance letter]

18 Jul 2024

PONE-D-24-10391R1 

PLOS ONE

Dear Dr. Oosterhof, 

I'm pleased to inform you that your manuscript has been deemed suitable for publication in PLOS ONE. Congratulations! Your manuscript is now being handed over to our production team.

Kind regards, 

on behalf of

Dr. Carmen María González-Domenech 

Academic Editor

PLOS ONE